# Using machine learning to forecast peak health care service demand in real-time during the 2022–23 winter season: A pilot in England, UK

**Roger A. Morbey**[1]*, **Dan Todkill**[1,2], **Phil Moura**[3], **Liam Tollinton**[4], **Andre Charlett**[5], **Conall Watson**[6], **Alex J. Elliot**[1]

1 Real-time Syndromic Surveillance Team, Field Services, Health Protection Operations, UK Health Security Agency, Birmingham, United Kingdom, 2 Health Sciences, Warwick Medical School, University of Warwick, Coventry, United Kingdom, 3 Department of Health and Social Care, London, United Kingdom, 4 Health Analytics and Automation, Data Analytics and Surveillance, UK Health Security Agency, London, United Kingdom, 5 Statistics, Modelling and Economics Division, UK Health Security Agency, London, United Kingdom, 6 Immunisation and Vaccine Preventable Diseases Division, UK Health Security Agency, London, United Kingdom

* roger.morbey@ukhsa.gov.uk

## Abstract

During winter months, there is increased pressure on health care systems in temperature climates due to seasonal increases in respiratory illnesses. Providing real-time short-term forecasts of the demand for health care services helps managers plan their services. During the Winter of 2022–23 we piloted a new forecasting pipeline, using existing surveillance indicators which are sensitive to increases in respiratory syncytial virus (RSV). Indicators including telehealth cough calls and emergency department (ED) bronchiolitis attendances, both in children under 5 years. We utilised machine learning techniques to train and select models that would best forecast the timing and intensity of peaks up to 28 days ahead. Forecast uncertainty was modelled usings a novel generalised additive model for location, scale and shape (gamlss) approach which enabled prediction intervals to vary according to the level of the forecast activity. The winter of 2022–23 was atypical because the demand for health-care services in children was exceptionally high, due to RSV circulating in the community and increased concerns around invasive group A streptococcal (iGAS) infections. However, our short-term forecasts proved to be adaptive forecasting a new higher peak once the increasing demand due to iGAS started. Thus, we have demonstrated the utility of our approach, adding forecasts to existing surveillance systems.

## 1. Introduction

Healthcare services are prone to experiencing periods of high burden and demand for services ('pressures') during winter months each year. These pressures can lead to severe problems in delivering critical health services, During winter months, healthcare pressures are exacerbated

**Data Availability Statement:** All relevant data are within the paper and its Supporting information files.

**Funding:** This study was funded by the National Institute for Health Research (NIHR). This work also benefits from the infrastructure and partnerships assembled by Health Data Research UK, including through the Data and Connectivity National Core Study, funded by UK Research and Innovation [grant reference MC_PC_20058].

**Competing interests:** The authors have declared that no competing interests exist.

by factors that can increase demand, including cold weather, respiratory pathogens, gastrointestinal pathogens and subsequent workforce absences [1]. In particular, the role of influenza and respiratory syncytial virus (RSV) in driving winter pressures has been extensively documented.

RSV is a major cause of bronchiolitis and bronchitis amongst young children [2] and although mainly produces mild symptoms, RSV infection can lead to severe illness in the immunocompromised [3] and is a major cause of death in infants globally [4].

During periods of heightened influenza and RSV activity, increases in demand can occur across a range of healthcare services from community physicians (general practitioners; GPs) through to specialist secondary care facilities. In England, RSV accounts for approximately 30,000 paediatric admissions in children aged <5 annually [5].

Identifying the key drivers underlying winter pressures is critical to understanding, managing and responding to the periods of high demand. Surveillance is a cornerstone of public health, monitoring changes in community-based activity of certain pathogens, diseases and conditions. Surveillance can provide a 'view' of key metrics that can be used to understand the drivers of pressures. Routinely collected surveillance data provide intelligence on those factors known to cause pressures e.g. monitoring increases in influenza cases. Surveillance data can also provide the opportunity to anticipate these pressures through predictions or forecasting.

Recently, advancements in machine learning have made it possible to develop more powerful and accurate forecasting models, utilising larger and more complex datasets. However, the key to developing accurate and timely models is the availability of suitable surveillance data that inform on healthcare service usage. Here, we use real-time syndromic surveillance data that are routinely collected as part of the UK Health Security Agency (UKHSA) public health surveillance programme to create short term forecasts for peak health care demand during periods of rising seasonal respiratory activity. We calculated forecast reliability to describe uncertainty around forecasts and piloted forecasts during the 2022–23 winter season and compared forecasts to actual activity.

## 2. Methods

We created two automated machine learning pipelines in R, firstly to select and train forecast models, secondly to create daily forecasts (Fig 1). We describe here the methods used following the flow of the pipelines.

### 2.1 Data selection

The UKHSA coordinates a programme of real-time syndromic surveillance that supports and augments other UKHSA health surveillance programmes [6]. The UKHSA real-time syndromic surveillance systems monitor anonymised health service contacts from across the National Health Service (NHS) in England. For this pilot study, we used two syndromic indicators that are routinely part of the ongoing UKHSA daily syndromic surveillance service; NHS 111 telehealth calls for 'cough' and emergency department (ED) attendances for 'acute bronchiolitis'. Both syndromic indicators were restricted to children aged five years or under because these indicators are known to be sensitive to seasonal outbreaks of RSV [7–9]. Using established indicators that are well understood aids interpretation and enables comparison with previous years.

We used anonymised health service data that are routinely used by UKHSA for public health surveillance of respiratory illnesses, including RSV. This study was part of ongoing work to improve the capabilities of UKHSA surveillance systems. As such, no specific approvals were required to use the anonymised data included this study.

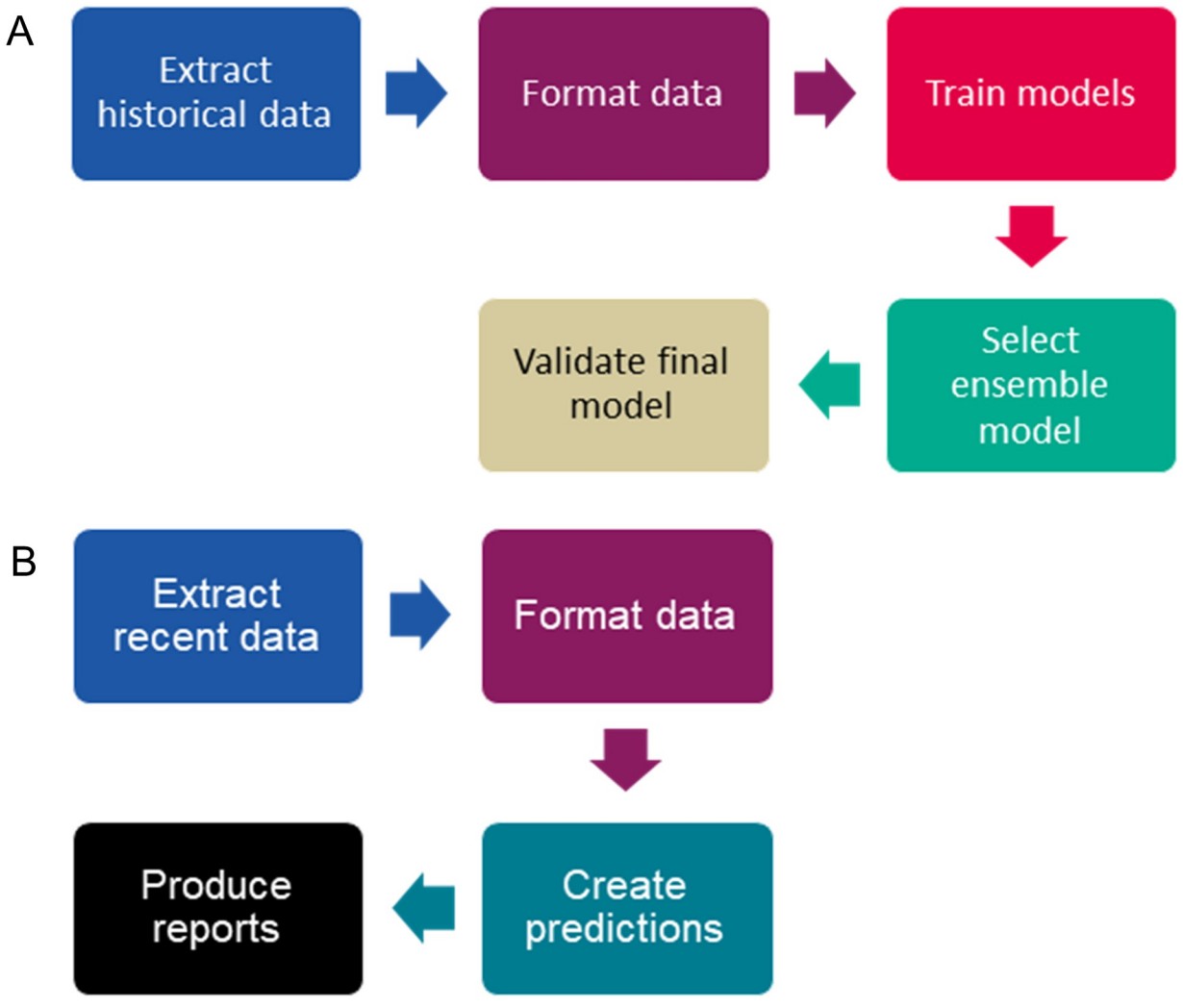

**Fig 1. Pipelines used for A) creating models and B) producing daily forecasts.**

## 2.2 Data cleaning and formatting

Firstly, data were smoothed to remove day of the week effects caused by weekends and public holidays [10]. A simple seven-day moving average would fail to account for the impacts of public holidays. Therefore, data were smoothed by weighting activity so that weekends and any public holidays within the week always contributed two sevenths of the weekly activity. Secondly data were normalised, so that the smoothed daily activity were in the range zero to one. Finally, derived variables were created for the forecast models (not all derived variables were used in all the models tested; Table 1) [11].

## 2.3 Training models

Rather than restrict forecasts to a single methodology, for both indicators we tested a wide range of alternate models, using the data to select the best method for each indicator [11]. Firstly, we choose seven alternate supervised machine learning methods; linear regression, generalised linear models with elastic net regularization (with and without internal

**Table 1. Derived variables used to create forecast models.**

| Derived variable | Descriptor |
| --- | --- |
| Trend slope | difference between today and yesterday |
| Rate of change | difference between today's slope and yesterdays |
| Seasonality | either month of year or Fourier transformations |
| Secular trend | linear or quadratic |
| Quadratic term | square of daily activity |
| Three-point-moving-average | Calculate for daily activity, trend slope and rate of change of slope |

optimisation of parameter lambda), k-Nearest-Neighbour regression, random forest regression, support vector machine for regression, and eXtreme Gradient Boosting regression.

We created four options for alternative models, in order to test which of our derived variables were most useful for producing accurate forecasts:

- Option 1: either no seasonality or including a day of month variable or a Fourier Transformation.

- Option 2: don't include a secular trend, or include a linear or a quadratic trend.

- Option 3: include the square of daily activity as an additional variable to enable the modelling or a quadratic rather than just linear relationship with daily activity.

- Option 4: to protect against the possibility of single-day spikes having an undue influence, use a 3 day moving average to replace the daily activity, trend and rate of change variables.

The combination of seven machine learning methods and the four options above gave (7 * 3 * 3 * 2 * 2) = 252 alternate 'model specifications' to be tested. Furthermore, for each model specification 28 models were created and trained to forecast from 1 to 28 days ahead respectively. Models were trained using historical data prior to October 2022, the supervised learning using actual data from either 1 to 28-days ahead of daily forecasts as the 'labels' for the target forecast. Historical data were split randomly into training and test data sets, 80% of the historical data being used for training.

## 2.4 Selection of ensemble model

For each of the 252 alternate model specifications an ensemble forecast was created which forecast when and how high activity was going to peak over the next 28 days. The 'forecast peak' for each model specification was defined as the highest value forecast by the individual 1 to 28-day ahead forecast models. Using the test data set, the forecast peaks were compared with the actual highest value or 'peak' that occurred in the 28 days following the forecast. An 'intensity error' was calculated as the difference between the height of the forecast peak and the actual peak. Similarly, a 'timing error' was calculated which was the difference in days between the day when activity peaked and the date when activity was forecast to peak. The intensity and timing errors were combined to give a single 'forecast peak error'. The forecast peak error includes weighting to emphasise the importance of accurately forecasting peaks when activity is high [12].

The peak error measure can be described as

$$y_d = \frac{\max(f_d, x_d)}{\max(x)} * \left( \left( 1 + \frac{t_d}{27} \right) * \left( 1 + \frac{i_d}{\max(i)} \right) - 1 \right) / 3$$

where $y_d$ is the peak error on day $d$, $x_d$ is the actual smoothed count on day $d$, $f_d$ is the forecast peak intensity on day $d$, $max(x)$ is the maximum of all actual and predicted smoothed counts, $t_d$ is the difference in days between the date of the peak forecast and when the actual peak occurred, $i_d$ is the difference between the peak forecast's intensity and the actual peak, and $max(i)$ is the maximum error seen in predicting forecast intensity. The peak error measure is zero if the peak forecast correctly identifies both the date and intensity of the peak. The measure increases as the difference between peak forecast intensity and actual peak intensity increases. Also, the measure increases as the difference between the forecast date and actual date of peak increases, but this increase is less if both actual and forecast activity is low.

The model specification which resulted in the smallest mean forecast peak error was selected for daily forecasts.

## 2.5 Model validation

Once the best model specification has been selected based on the training data, forecasts were retrained using all the available historical data. The historical intensity and timing errors were calculated for forecast peaks and used to estimate forecast uncertainty. To allow for variation in the standard deviation of errors as activity approaches a peak, a generalised additive model for location, scale and shape (gamlss) model was used to estimate standard deviation variation against the level of current activity [13]. Thus, we created uncertainty intervals which could vary as activity approaches a seasonal peak.

## 2.6 Creating daily reports

To produce daily forecasts, recent data are extracted and formatted using the same data processes as for training models. The validated model for each indicator is used to produce 1 to 28-day ahead forecasts based on the latest data available. These forecasts are used to create daily reports which predict when activity will peak in the next 28 days and at what level, including the uncertainty intervals.

## 3. Results

### 3.1 Model selection

The model specifications with the lowest forecast peak errors for both indicators used a random forest learning method, with seasonality modelled by Fourier transformations. However, the other specification options differed between the two indicators. The lowest forecast peak errors for the NHS 111 cough calls data included a quadratic term for activity, a quadratic secular trend and averaging over three consecutive days' data points. Whilst for ED acute bronchiolitis attendances the lowest errors involved a linear trend and no quadratic term for activity or averaging over consecutive days. S1 Table shows the forecast peak errors for each model specification. In general, errors were lower for NHS 111 calls than for ED bronchiolitis attendances, with 52 NHS 111 model specifications performing better than the best ED specification. Overall, including seasonality improved peak forecasts, with Fourier transformations performing better than seasonality using months. The learning method with the lowest mean errors was random forest, followed by k-nearest neighbour regression (Table 2).

### 3.2 Model validation

The gamlss models show that the variation in intensity errors increase as actual counts increase (Table 3). By contrast, the variation in timing errors decrease as actual counts increase.

**Table 2.  Mean forecast peak errors by method and syndromic surveillance system.**

| Learning method | ED[1] attendances | NHS 111 calls[2] |
|---|---|---|
| random forest | 0.0267 | 0.0195 |
| k-nearest-neighbour | 0.0359 | 0.0239 |
| support vector machine | 0.0399 | 0.0287 |
| glm with elastic net regularization including optimised lambda | 0.0483 | 0.0355 |
| linear regression | 0.0490 | 0.0378 |
| generalised linear model (glm) with elastic net regularization | 0.0487 | 0.0418 |
| extreme gradient boosting | ^0.1483 | 0.1307 |

^Only 6 out of 36 model specifications converged.

[1]Emergency department;

[2]National Health Service 111

**Table 3.  Gamlss model coefficients for variation in error standard deviation vs actual counts.**

| Error type | System | Intercept | Coefficient |
|---|---|---|---|
| Intensity | NHS 111 | 24.5 | 1.0006 |
|  | EDSSS | 18.9 | 1.0055 |
| Timing | NHS 111 | 5.9 | 0.9994 |
|  | EDSSS | 12.0 | 0.9998 |

The gamlss model coefficients were used to create confidence intervals around the timing and intensity errors, which varied depending on the number of actual counts at the time of the forecast (Figs 2 and 3). Intensity errors are not symmetric because forecast peaks cannot be negative, thus when actual counts are low a forecast peak can over-estimate by more than it can under-estimate.

### 3.3 Pilot season 2022–23

During October 2022, ED acute bronchiolitis attendances in children aged under 5 years reporting to EDSSS increased until a peak of 220.0 attendances on 31 October. Subsequently, there was a decrease until 5 November before attendances increased again reaching a seasonal high of 311.4 attendances on 29 November. Similarly, NHS 111 calls for cough in children under 5 years rose to a peak of 991.6 calls on 22 October, decreased until 2 November and then started rising. However, whilst the increase in NHS 111 calls slowed prior to 30 November it was then followed a sharp increase in calls, reaching a seasonal high of 1,842.9 calls on 6 December 2022.

The seasonal peak in ED attendances at the end of November coincided with the usual timing of peak RSV activity seen in previous years (as monitored by laboratory reporting) [14]. The additional increase in NHS 111 calls after 30 November 2022 was unprecedented, being 39.3% higher than the previous highest winter peak, 1,323.4 on 7 December 2019. Consequently, the level of activity was outside the range of anything seen in the training data.

The pilot forecast made on 26 November 2022, forecast that ED attendances for acute bronchiolitis would peak at 340.1 attendances on 29 November (Fig 4). The same day forecast for NHS 111 calls predicted that they had already peaked. The timing for the ED forecast was correct but the level of the peak was an over-estimate of 29.3 (9.4%) attendances.

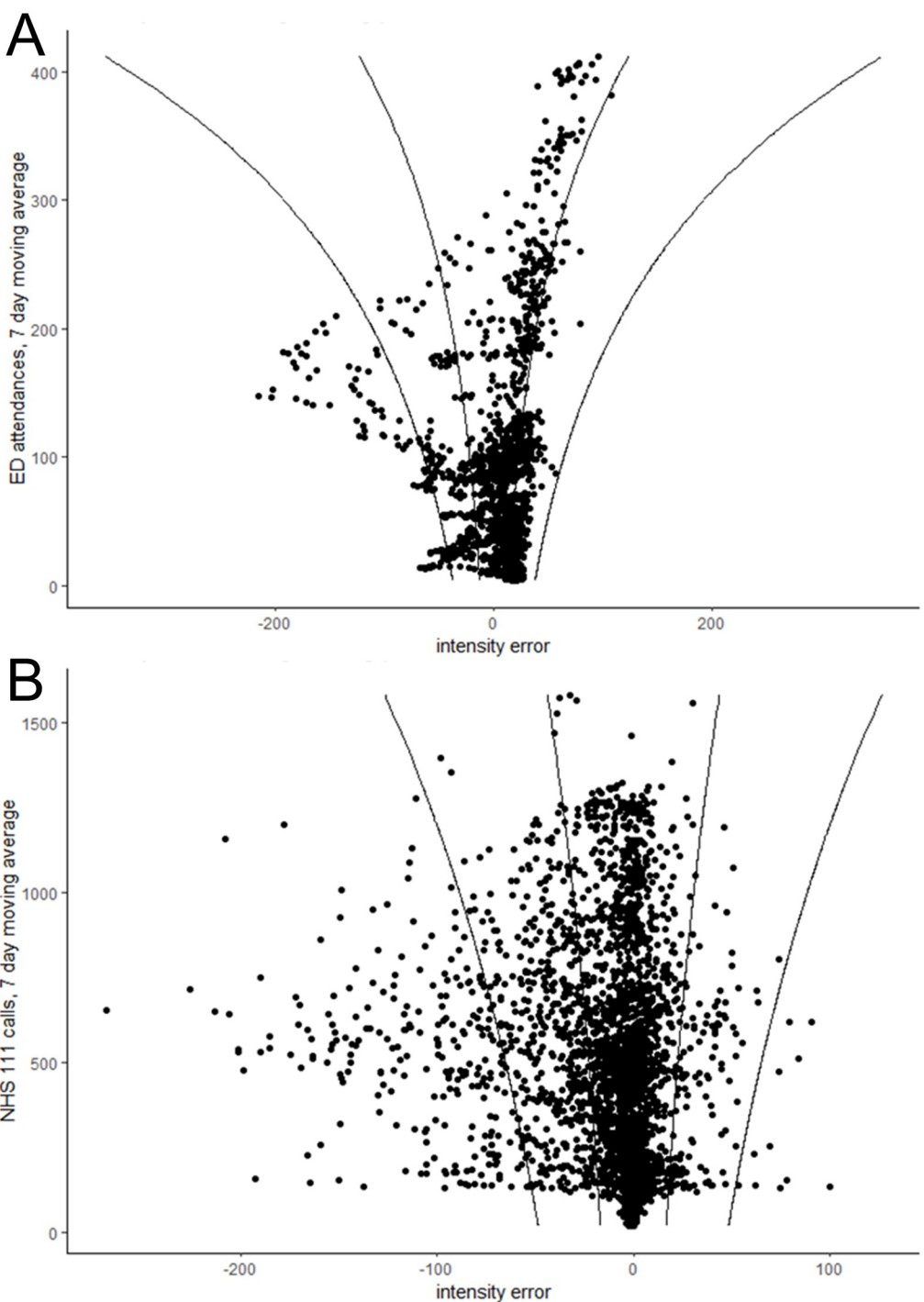

**Fig 2. Intensity errors during training period for A) emergency department acute bronchiolitis attendances and B) NHS 111 cough calls.** Lines show 50% and 95% confidence intervals.

The NHS 111 forecasts failed to predict the unprecedented rise in NHS 111 cough calls in children in December 2022 until the rise had started. However, a forecast using a linear regression learning method proved to be adaptive, forecasting a later and higher seasonal peak once activity began to rise sharply at the start of December 2022 (Fig 5).

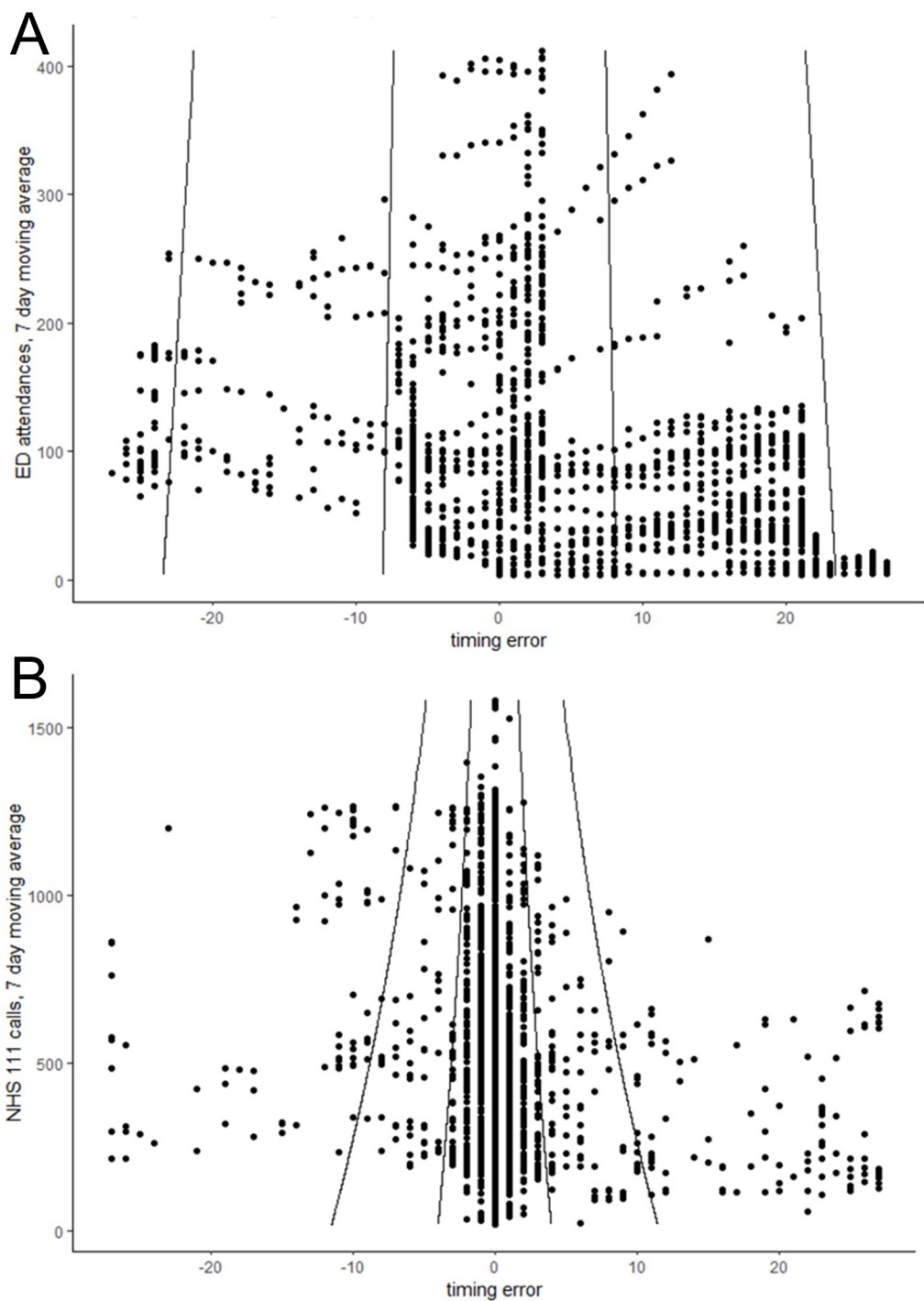

**Fig 3. Timing errors during training period for A) emergency department acute bronchiolitis attendances and B) NHS 111 cough calls.** Lines show 50% and 95% confidence intervals.

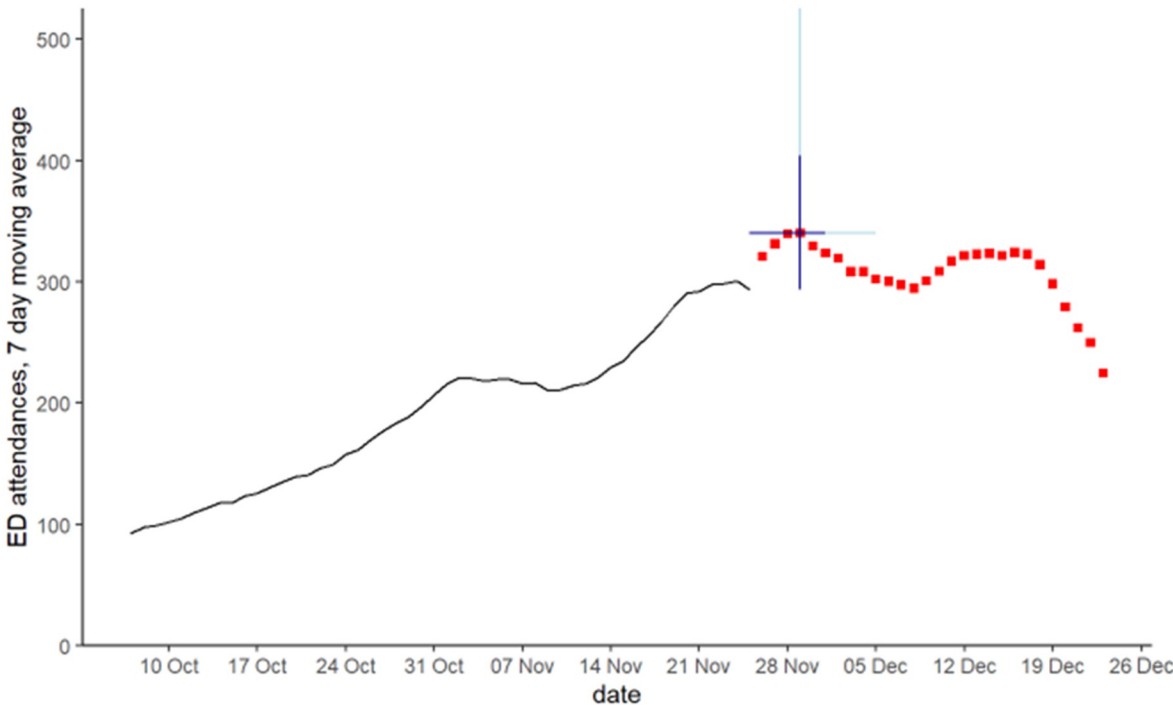

**Fig 4. 28 day forecast for peak emergency department acute bronchiolitis attendances in children aged <5 years.** Red squares are 28 day forecast, blue lines show 50% (dark blue) and 95% (light blue) data intervals around the peak forecast.

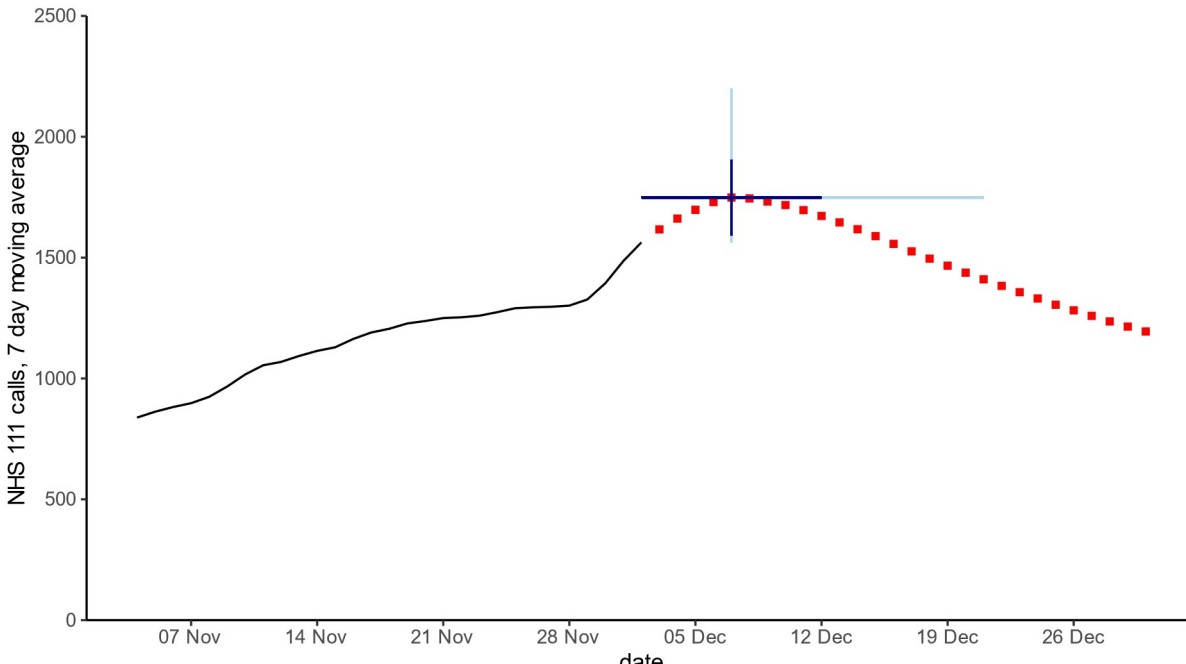

**Fig 5. 28 day forecast for peak NHS 111 cough calls in children aged <5 years.** Red squares are 28 day forecast, blue lines show 50% (dark blue) and 95% (light blue) data intervals around the peak forecast.

## 4. Discussion

### 4.1 Key findings

Machine learning pipelines can be used to train, select models and create daily forecast reports that predict the peak in demand for RSV activity over the following 28 days. During 2022 our pilot forecasts were able correctly predict the peak in ED acute bronchiolitis attendances in children under 5 years. During November 2022, our forecasts for NHS 111 cough calls in children aged under 5 years, predicted a similar peak was going to occur as in previous pre-pandemic years. However, as cough calls began to increase sharply at the start of December, our forecasts also began to change, predicting a later peak in December that was higher than previous years.

### 4.2 What was known before

Prior to the COVID-19 pandemic, seasonal activity for RSV was predictable in its seasonality, peaking in England at the end of November to beginning of December [15–17]. However, during 2020 and 2021, the seasonality of RSV was disrupted with no winter peak in 2020 and a deferred peak occurring in summer 2021 [7]. Traditional surveillance methods based on historical data and recurring seasonality [18] continued to predict syndromic indicators would rise in winter 2020 due to RSV. Whilst short-term forecasts based on recent trends are more adaptive, these too would not perform well during atypical seasons, unless seasonality was excluded from model variables [11].

### 4.3 Interpretation of findings

The unexpected dramatic rise in NHS 111 cough calls in December 2022 coincided with unusual increases in group A streptococcus (GAS) infections in children [19,20]. This unusual seasonal activity combined with the impact of media reporting [21] resulted in an unprecedented increase in NHS 111 calls in children, particularly those indicators linked to symptoms related to GAS infections e.g. sore throat, fever and cough. However, ED attendances were less affected by changes in patient presenting behaviour, and consequently ED syndromic indicators did not have an additional large peak. The differences between syndromic systems illustrate some of the strengths and weaknesses of syndromic surveillance. Syndromic indicators cannot identify specific causal pathogens, thus NHS 111 cough calls although sensitive to RSV are not specific enough to exclude other causal factors. Thus, NHS 111 cough calls were not a reliable indicator for assessing the total burden of health care demand attributable to RSV. However, if policy and decision-makers need to understand the current pressures on health services from all causes then syndromic indicators are more sensitive than pathogen-specific surveillance such as laboratory reporting. Importantly, a syndromic surveillance service that comprises a range of data sources across the spectrum of health services is better able to distinguish between pressures due to changes in underlying disease incidence and those due to changing patient behaviour.

### 4.4 Limitations

Inevitably, forecasts trained on historical data will perform better when current data are within the range and seasonality seen previously. Previously, we have shown that a model that is not trained to expect a recurring seasonal pattern performs better during atypical seasons [11]. However, the unexpected peak in December 2022 NHS 111 calls was not out of season and so the accuracy of forecasts was not due to inclusion of seasonality variables. In this case, we found that some of the regression methods which performed best using our forecast peak

error measure, e.g. random forest, generated forecasts that assumed activity had already peaked. By contrast, using simple linear regression generated forecasts that correctly predicted activity was going to continue to increase in line with current trends. Therefore, the simpler method outperformed the method automatically selected by our machine learning algorithm.

The unprecedented increase in NHS 111 cough calls in children during December 2022 revealed a limitation in the use of this indicator for forecasting peak pressures due to RSV. The exceptional additional winter pressures were not due to RSV but due to reaction to GAS media reports. Thus, without additional intelligence, a report intended to show pressures due to RSV could have misinterpreted as showing that RSV activity was exceptionally high.

## 4.5 Comparison with existing approaches

Previous research into machine learning forecasts have, similarly, compared a number of different model methods [22–24]. For example, Do et al. compared the mean square errors for 16 different model methods [22]. Whilst Do et al. studied diarrhoeal disease, Castro Blanco et al. considered the onset of the winter influenza season in Spain, they compared random forest, support vector machine and logistic regression, finding logistic regression to be the most accurate [23].

Machine learning methods have been used for forecasting where the analysis is complicated due to the large number of predictive variables [22–24]. By contrast, we have deliberately restricted our predictive variables to the syndromic indicators readily available during ongoing daily surveillance. Similarly our approach did not require mechanistic assumptions about the development of infectious diseases, whilst these approaches can provide accurate forecasts they require considerably more work to develop [25]. Our approach is in line with a US study into 22 rival influenza forecasting methods which found that the timeliness of reporting and integration into real-time public health decision-making are as important as forecast accuracy [26].

Our approach was to compare different machine learning models and use the method that best predicted our data, an alternative approach used elsewhere is to take a weighted average of all models [22]. Whilst these 'ensemble forecast methods' inevitably fit the data better, they lack transparency, and it is difficult to explain the theoretical justification for individual forecasts.

## 4.6 Public health implications

Short-term forecasts can provide additional information compared to existing surveillance baselines based on previous years because they are more adaptive to recent changes in trends. Our automated pipeline for creating short-term forecasts of seasonal peaks is useful in identifying the timing and intensity during typical seasons. Furthermore, the process of using machine learning methods to produce a reproducible automated pipeline means new indicators can easily and quickly be added to syndromic reports. However, automated reporting of trends and forecasts for syndromic data should always be accompanied with expert interpretations which can warn of emerging events. For instance, where a real-time change in patient behaviour means one or more indicators is no longer comparable with previous years. Improved automation and real-time interpretation are important as we may need to create forecasts quickly when notified that there is an increase in disease incidence. Also, the same pipeline can be used to assess other causal pathogens, including influenza and SARS-Cov2. Similarly, it may be possible to model non-infectious diseases such as allergic rhinitis (hay fever) where the historical data includes recurring seasonal peaks.

### 4.7 Recommendations and future work

Importantly, when current data are outside the range of training data, or seasonality does not match the training data forecasts should be interpreted in caution. We'd recommend that any forecasts used for routine surveillance include tests for data that are outside the range of testing data. Also, we suggest that when increases start to occur out-of-season forecast models are selected that do not include seasonality variables.

In future, it may be possible to provide better forecasts during atypical seasons by weighting the training data to give more emphasis to the rare events. Also, synthetic data could be incorporated in training data to allow for plausible events that have not yet occurred in the training data, e.g. out-of-season outbreaks, or more virulent pathogens.

## Supporting information

**S1 Table. Forecast peak errors for each model specification.**
(DOCX)

## Acknowledgments

The authors would like to thank the syndromic data providers including: NHS 111 and NHS England (NHS 111 telehealth), and emergency department clinicians and NHS Trusts and NHS England supporting emergency department syndromic surveillance. Roger A. Morbey, Dan Todkill and Alex J Elliot are affiliated with the NIHR Health Protection Research Unit (HPRU) in Emergency Preparedness and Response at King's College London. Alex J Elliot is affiliated with the NIHR HPRU in Gastrointestinal Infections at University of Liverpool. Dan Todkill is supported by the NIHR Applied Research Collaboration (ARC) West Midlands. The views expressed are those of the author(s) and not necessarily those of the NIHR, the UK Health Security Agency or the Department of Health and Social Care.

## Author Contributions

**Conceptualization:** Roger A. Morbey, Dan Todkill, Phil Moura, Alex J. Elliot.

**Data curation:** Roger A. Morbey.

**Formal analysis:** Roger A. Morbey.

**Funding acquisition:** Dan Todkill.

**Investigation:** Roger A. Morbey.

**Methodology:** Roger A. Morbey, Phil Moura.

**Project administration:** Roger A. Morbey, Dan Todkill.

**Software:** Roger A. Morbey, Phil Moura, Liam Tollinton.

**Supervision:** Alex J. Elliot.

**Validation:** Liam Tollinton, Andre Charlett, Conall Watson, Alex J. Elliot.

**Writing – original draft:** Roger A. Morbey.

**Writing – review & editing:** Roger A. Morbey, Dan Todkill, Phil Moura, Liam Tollinton, Andre Charlett, Conall Watson, Alex J. Elliot.

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
