## [Decision Letter · Decision Letter 0]

14 Nov 2024

PONE-D-23-30068Using machine learning to forecast peak health care service demand in real-time during the 2022-23 winter season: a pilot in England, UKPLOS ONE

Dear Dr. Morbey,

Thank you for submitting your manuscript to PLOS ONE. After careful consideration, we feel that it has merit but does not fully meet PLOS ONE’s publication criteria as it currently stands. Therefore, we invite you to submit a revised version of the manuscript that addresses the points raised during the review process.

We look forward to receiving your revised manuscript.

Kind regards,

Yury E Khudyakov, PhD

Academic Editor

PLOS ONE

Journal Requirements:

3. Thank you for stating the following financial disclosure: “This study was funded by the National Institute for Health Research (NIHR). This work also benefits from the infrastructure and partnerships assembled by Health Data Research UK, including through the Data and Connectivity National Core Study, funded by UK Research and Innovation [grant reference MC_PC_20058].”

4. We note that you have indicated that there are restrictions to data sharing for this study. PLOS only allows data to be available upon request if there are legal or ethical restrictions on sharing data publicly. For more information on unacceptable data access restrictions, please see http://journals.plos.org/plosone/s/data-availability#loc-unacceptable-data-access-restrictions. Before we proceed with your manuscript, please address the following prompts: a) If there are ethical or legal restrictions on sharing a de-identified data set, please explain them in detail (e.g., data contain potentially identifying or sensitive patient information, data are owned by a third-party organization, etc.) and who has imposed them (e.g., a Research Ethics Committee or Institutional Review Board, etc.). Please also provide contact information for a data access committee, ethics committee, or other institutional body to which data requests may be sent. b) If there are no restrictions, please upload the minimal anonymized data set necessary to replicate your study findings to a stable, public repository and provide us with the relevant URLs, DOIs, or accession numbers. For a list of recommended repositories, please see https://journals.plos.org/plosone/s/recommended-repositories. You also have the option of uploading the data as Supporting Information files, but we would recommend depositing data directly to a data repository if possible. We will update your Data Availability statement on your behalf to reflect the information you provide.

5. Please remove your figures from within your manuscript file, leaving only the individual TIFF/EPS image files, uploaded separately. These will be automatically included in the reviewers’ PDF**.**

Additional Editor Comments:

Your manuscript was reviewed by three experts in the field. The reviewers identified several important problems in your submission. Please review the attached comments and provide point-by-point responses.

Reviewers' comments:

Reviewer's Responses to Questions

**Comments to the Author**

1. Is the manuscript technically sound, and do the data support the conclusions?

Reviewer #1: Yes

2. Has the statistical analysis been performed appropriately and rigorously? 

Reviewer #1: Yes

3. Have the authors made all data underlying the findings in their manuscript fully available?

Reviewer #1: Yes

4. Is the manuscript presented in an intelligible fashion and written in standard English?

Reviewer #1: Yes

5. Review Comments to the Author

Reviewer #1: The authors piloted a new forecasting pipeline during the Winter of 2022-23, using existing surveillance indicators which are sensitive to increases in respiratory syncytial virus (RSV). Indicators including telehealth cough calls and ED bronchiolitis attendances, both in children under 5 years. They utilised machine learning techniques to train and select models that would best forecast the timing and intensity of peaks up to 28 days ahead. Forecast uncertainty was modelled usings a novel gamlss approach which enabled prediction intervals to vary according to the level of the forecast activity.

They calculated forecast reliability to describe uncertainty around forecasts and piloted forecasts during the 2022-23 winter season and compared forecasts to actual activity.

They have demonstrated the utility of the approach, adding forecasts to existing surveillance systems.

6. PLOS authors have the option to publish the peer review history of their article (what does this mean?). If published, this will include your full peer review and any attached files.

Reviewer #1: No

---

## [Author Response · Author response to Decision Letter 0]

23 Dec 2024

Response to reviewers

Reviewer #1

The authors piloted a new forecasting pipeline during the Winter of 2022-23, using existing surveillance indicators which are sensitive to increases in respiratory syncytial virus (RSV). Indicators including telehealth cough calls and ED bronchiolitis attendances, both in children under 5 years. They utilised machine learning techniques to train and select models that would best forecast the timing and intensity of peaks up to 28 days ahead. Forecast uncertainty was modelled usings a novel gamlss approach which enabled prediction intervals to vary according to the level of the forecast activity.

They calculated forecast reliability to describe uncertainty around forecasts and piloted forecasts during the 2022-23 winter season and compared forecasts to actual activity.

They have demonstrated the utility of the approach, adding forecasts to existing surveillance systems.

Response: we thank the reviewer their comments. 

Reviewer #2

Authors proposed "Using machine learning to forecast peak health care service demand in real-time during the 2022-23 winter season: a pilot in England, UK". The structure of the article is well structured but authors should consider the following comments.

1. Proofread the entire manuscript.

2. Draw a graphical abstract of your proposed approach

3. compare your approach with previous existing approaches

Response: We have proofread the manuscript and included the formatting changes requested by the editor to ensure compliance with the journal style. Figure 1 provides an overview of the processes involved in our forecasting; we are happy for this to be used as a graphical abstract if required by the journal format. 

We have added a new section to the discussion (4.5 Comparison with existing approaches) which now provides a comparison with existing forecasting approaches. We have supported this new discussion with 5 new references. 

Reviewer #3

a) Language

Minor language errors such as "Derived variables were created that were used to create the forecast models."

Response: We have made corrections to improve the grammar and readability of the paper.

Some abbreviations used before or without definition: gamlss.

Response: We have spelt out gamlss and included a reference. We have also checked throughout the manuscript for proper formatting of other abbreviations. 

b) Methods

The methods should be more thoroughly explained and the sections should better structured.

Response: We have expanded the methods section as explained below, including making the structure of the model specifications clearer through the combination of different machine learning methods and model options for which derived variables to include.

The current state of the paper does not permit reproducing the results or even applying the proposed solution to other data, because of the lack of details in what the methods are concerned. Some examples:

- why were only two syndromic indicators used. Did the authors try others as well? What is the justification for choosing those two?

Response: We will make the data used in our analysis publicly available. We can also make the scripts we used for the analysis available to enable others to reproduce our results or apply the same methods to other data.

In this pilot forecast we focussed on forecasting RSV using two of the most established indicators available in our routine surveillance. However, we have developed our pipelines and methodology so it will be easy to extend this work to other indicators and it is our intention to consider influenza-like-illness and other respiratory indicators in future work.

- the authors state that they used a smoothing method to account for the effects of holidays and weekend and simply reference an article without giving any other detail. The paper should be self-contained, meaning they need to at least mention what kind of smoothing method they use, and, possible particularities with respect to their data, if any.

Response: We have added more detail about our approach to account for day of the week effects.

- the authors state: "For each of these seven different methods we included the following options:" and proceed to enumerate 4 such options. These options are in fact data preprocessing methods - as far as I can tell, should be listed as such and should be better explained. This part requires serious review.

Response: The reviewer is correct that these options are introduced part of the preprocessing. However, they do create distinct models, for instance whether or not seasonality is included in a model. We have expanded this section to make clearer what we mean by different methods and different models.

What does "With or without a quadratic term for current activity" mean? To what current activity is this referring to and how exactly (formally) is this used?

Response: We have expanded our explanation of the derived variables and the options we used. We used an optional additional variable that was the square of daily activity so that we could compare models that had a linear or a quadratic relationship with daily activity.

- the authors first describe the data, then the methods, then they come back to describing the data (see, for example, paragraph starting with "We used anonymised health service data that is routinely used by UKHSA for public health surveillance of respiratory illnesses, including RSV."). The section should be structured for better readability.

Response: Thank you for this suggestion we have moved this sentence to section 2.1 on data selection.

- "The forecast peak error includes weighting to emphasise the importance of accurately forecasting peaks when activity is high." What values were used for the weights? This is an important detail that should also be motivated, let alone mentioned: is it 0.5 for both? Why?

Response: Apologies for not including this important measure, which we had published in a previous paper. We have now reproduced the full details on how this is calculated along with the original citation.

- "a gamlss model was used to estimate standard deviation variation against the level of current activity." What kind of distribution.

Response: We have included the citation for the original work on gamlss.

[end]

---

## [Decision Letter · Decision Letter 1]

15 Jan 2025

Using machine learning to forecast peak health care service demand in real-time during the 2022-23 winter season: a pilot in England, UK

PONE-D-23-30068R1

Dear Dr. Morbey,

We’re pleased to inform you that your manuscript has been judged scientifically suitable for publication and will be formally accepted for publication once it meets all outstanding technical requirements.

Kind regards,

Yury E Khudyakov, PhD

Academic Editor

PLOS ONE

Additional Editor Comments (optional):

Reviewers' comments:

Reviewer's Responses to Questions

**Comments to the Author**

1. If the authors have adequately addressed your comments raised in a previous round of review and you feel that this manuscript is now acceptable for publication, you may indicate that here to bypass the “Comments to the Author” section, enter your conflict of interest statement in the “Confidential to Editor” section, and submit your "Accept" recommendation.

Reviewer #3: All comments have been addressed

2. Is the manuscript technically sound, and do the data support the conclusions?

Reviewer #3: (No Response)

3. Has the statistical analysis been performed appropriately and rigorously? 

Reviewer #3: (No Response)

4. Have the authors made all data underlying the findings in their manuscript fully available?

Reviewer #3: Yes

5. Is the manuscript presented in an intelligible fashion and written in standard English?

Reviewer #3: (No Response)

6. Review Comments to the Author

Reviewer #3: (No Response)

7. PLOS authors have the option to publish the peer review history of their article (what does this mean?). If published, this will include your full peer review and any attached files.

Reviewer #3: No

---

## [Editor Report · Acceptance letter]

17 Jan 2025

PONE-D-23-30068R1 

PLOS ONE

Dear Dr. Morbey, 

I'm pleased to inform you that your manuscript has been deemed suitable for publication in PLOS ONE. Congratulations! Your manuscript is now being handed over to our production team.

Kind regards, 

on behalf of

Dr. Yury E Khudyakov 

Academic Editor

PLOS ONE